# Design and Testing of Augmented Reality-Based Fluorescence Imaging Goggle for Intraoperative Imaging-Guided Surgery

**DOI:** 10.3390/diagnostics11060927

**Published:** 2021-05-21

**Authors:** Seung Hyun Lee, Yu Hua Quan, Min Sub Kim, Ki Hyeok Kwon, Byeong Hyeon Choi, Hyun Koo Kim, Beop-Min Kim

**Affiliations:** 1Institute of Global Health Technology, College of Health Science, Korea University, Seoul 02841, Korea; aksska82@korea.ac.kr; 2Department of Biomedical Sciences, College of Medicine, Korea University, Seoul 02841, Korea; hwa1983418@gmail.com (Y.H.Q.); baby2music@gmail.com (B.H.C.); 3Department of Thoracic and Cardiovascular Surgery, Korea University Guro Hospital, College of Medicine, Korea University, Seoul 08308, Korea; 4Department of Bio-convergence Engineering, College of Health Science, Korea University, Seoul 02841, Korea; kimminseob@korea.ac.kr; 5Department of Interdisciplinary Bio/Micro System Technology, College of Engineering, Korea University, Seoul 02841, Korea; tomorrow6321@korea.ac.kr; 6Department of Bioengineering, College of Health Science, Korea University, Seoul 02841, Korea; 7Interdisciplinary Program in Precision Public Health, Korea University, Seoul 02841, Korea

**Keywords:** augmented reality, fluorescence image, neuro-navigation, smart goggle

## Abstract

The different pathways between the position of a near-infrared camera and the user’s eye limit the use of existing near-infrared fluorescence imaging systems for tumor margin assessments. By utilizing an optical system that precisely matches the near-infrared fluorescence image and the optical path of visible light, we developed an augmented reality (AR)-based fluorescence imaging system that provides users with a fluorescence image that matches the real-field, without requiring any additional algorithms. Commercial smart glasses, dichroic beam splitters, mirrors, and custom near-infrared cameras were employed to develop the proposed system, and each mount was designed and utilized. After its performance was assessed in the laboratory, preclinical experiments involving tumor detection and lung lobectomy in mice and rabbits by using indocyanine green (ICG) were conducted. The results showed that the proposed system provided a stable image of fluorescence that matched the actual site. In addition, preclinical experiments confirmed that the proposed system could be used to detect tumors using ICG and evaluate lung lobectomies. The AR-based intraoperative smart goggle system could detect fluorescence images for tumor margin assessments in animal models, without disrupting the surgical workflow in an operating room. Additionally, it was confirmed that, even when the system itself was distorted when worn, the fluorescence image consistently matched the actual site.

## 1. Introduction

During cancer surgery, although preoperative imaging techniques such as computed tomography (CT) and positron emission tomography (PET) have a meaningful impact on preoperative planning, the surgeon’s eyes and hands remain the decisive factors [1,2,3]. It can be difficult discriminating between malignant and normal tissues during actual clinical practice [4,5]. This may result in incomplete resection or unnecessary normal tissue resection [6]. Recently, indocyanine green (ICG)-based near-infrared (NIR) fluorescence imaging technology has been actively used to observe various types of cancer [7,8,9,10,11] and lymph nodes [12,13], as well as important structures, blood vessel formation [14,15], and blood perfusion [16], in real time during surgery. Furthermore, in previous studies, our group has employed ICG for applications such as sentinel lymph node (SLN) detection [17,18,19], assessment of lung segments [14], gastric conduit perfusion [20], and lung cancer detection [10]. However, as these systems display information on a remote monitor, surgeons are required to look at the monitor in order to identify the NIR fluorescence image [21,22,23]. This interrupts the surgeons’ attention, thereby increasing the probability of errors and the overall surgery time [24,25].

To overcome the abovementioned limitations, intraoperative systems with a head mount display (HMD) were developed (Table 1). Most existing HMDs are binocular and provide an immersive viewing experience [26]. Video see-through HMDs use a small display to show images from a computer or a camera and thus, create an immersive and virtual reality environment. However, the lack of direct visual access to the scenes interferes with the surgeons’ ability to assess and resect tissue spontaneously during surgery. Optical see-through HMDs are based on augmented reality (AR); in these devices, see-through displays are used to project images directly in the user’s field-of-view (FOV) [27,28,29], creating an AR environment [26,30]. Using this system, the user can view both the projected image and the object itself. Additionally, it supports visual tissue assessment and resection [31]. However, previous optical see-through systems are composed of optoelectrical components, making them bulky and difficult to wear. These features reduce the surgeon’s manipulability and block the surgeon’s vision. Furthermore, different pathways exist between the camera and the eye position. Therefore, it was necessary to use additional components or algorithms to match the user’s view field and the fluorescence image from the camera.

In this study, we developed an augmented reality-based fluorescence imaging (ARFI) goggle system that synchronizes the user’s view and the NIR camera’s view, in order to decrease related costs and improve the efficiency of the system. To assess its clinical applicability, the developed system was tested through preclinical experiments using a cancer model for guiding tumor resection in mice and rabbits.

## 2. Materials and Methods

### 2.1. Hardware Design for the Arfi System

The prototype of the ARFI system is presented in Figure 1. The proposed ARFI system consists of a customized NIR camera with the OS05A20 sensor (CMOS, Omnivision, Santa Clara, CA, USA) for fluorescence imaging, a commercial smart glass (Moverio BT-300, Epson, Suwa-shi, Nagano, Japan) for real-time display of the fusion image, and an optical system for utilizing the smart glass’s curved mirror. The CMOS camera is used to acquire a fluorescence image with a resolution of 2688 × 1944 pixels and a frame rate of 60 frames per second. The ARFI system was constructed by attaching the NIR camera onto the smart glass hardware, as shown in Figure 1. To acquire the fluorescence images, an 814–870 nm bandpass filter (FF01-842/56-25, Semrock, Rochester, NY, USA) was attached in front of the NIR camera to block all incoming light, except for the fluorescence emission from ICG. The smart glass system was equipped with two display panels (0.43˝ wide panel), each with a resolution of 1280 × 720 pixels. After processing the acquired images, the surgical and fluorescence images could be merged on the display. The optical system comprised a 20 mm square dichroic beam splitter (DBS, **#**62-628, Edmund Optics, Barrington, NJ, USA), a 20 mm protected gold coated 90° specialty mirror (#65-849, Edmund Optics, Barrington, NJ, USA), and mounts (DBS, mirror, and NIR camera mounts). Each mount was modeled using a 3D modeling program (Fusion 360, Autodesk, San Rafael, CA, USA) and printed with a 3D printer. These mounts could shift easily because they were connected via rods (SR6, SR3-P4, Thorlabs, Newton, NJ, USA). To co-register a fluorescence image with a surgical scene image, the DBS mount employed a small spring, screws, and a ball to rotate along the three axes; it could be adjusted horizontally to match the position of the user’s eye. The NIR camera mount was grooved to allow for heat release, and it could be adjusted vertically. The central mirror mount secured the triangular mirror and the smart glass; it also linked the other mounts via the rod, as shown in Figure 1. Owing to these mounts, the system could be customized depending on the user; this increased the accuracy of the fluorescence image registration.

### 2.2. Operational Principle of Arfi System

The schematic of the ARFI system is shown in Figure 2a. The excitation light source for ICG was used by attaching a 790 nm short-pass filter (FF01-790/SP-25; Semrock, Rochester, NY, USA) in front of an LED source (M780L3, Thorlabs, Newton, NJ, USA) with a 780 nm peak, and focused on the surgical field. This filter was used only to detect the fluorescence emitted by ICG on blocking light with a similar wavelength as the fluorescence reflected from the sample. White light from an array of four Luxeon III white light diodes was used (Lumileds, San Jose, CA, USA); all NIR wavelengths were filtered out from this light (E680SP, Chroma). The generated fluence rates for NIR excitation and white light were 0–5 mW/cm^2^ and 0–1 mW/cm^2^, respectively. The distance between the source and surgical field was approximately 250 mm. The DBS transmitted visible light enabling the surgeon to view the actual surgical scene, while also reflecting the near-infrared rays entering the same optical path as that of the surgeon. In this case, the near-infrared rays are reflected again through the mirror and then continue toward NIR cameras, such that the NIR camera could acquire fluorescence images. Thus, NIR images were acquired on the same optical path as the actual surgical scene through an optical separation system, and they were then presented to the surgeon. The major benefit of using two different images that share the same optical path is that easy blending between the NIR and actual images can be achieved.

As demonstrated in Figure 2b, the operation of the entire system is as follows. The typical working distance to acquire a focused fluorescence image is approximately 500 mm. Fluorescence images acquired from the NIR cameras were delivered to a computer through a USB 3.0 port and subjected to post-processing. Subsequently, they were transmitted to the smart glass via Miracast [34], a wireless image transmission technique, and were then projected to the eyes in real time. As the real-time projected image is overlaid on the actual field, users can obtain useful information, including the margin of cancer, in real time. The acquisition of fluorescence signals from NIR cameras and the post-processing were achieved using Visual Studio 2017 (Microsoft, Redmond, WA, USA), C++, and OpenCV library. Camera grabbing, which refers to image acquisition via NIR cameras, was performed using OpenCV library functions. Signals above a specific threshold value were saved from removing background noise from the raw fluorescent images. Then, to improve the contrast with the surrounding area, the NIR fluorescence image was pseudo-colored in green and overlaid with 100% transparency over the color video image of the same surgical field. During the surgery, the surgeon cannot realign the hardware immediately. Therefore, the software can be used to manually adjust the magnification, reduction, and rotation of the fluorescence image to match the actual field.

### 2.3. In Vivo Animal Studies

This study was approved by the Institutional Animal Care and Use Committee of Korea University College of Medicine (IACUC approval number: KOREA-2016-0228). Six-week-old C57BL/6 mice (20–25 g; Orient Biotech, Seongnam-si, Gyeonggi-do, Korea) and female New Zealand white rabbits (2.5–3 kg; DooYeol Biotech Co Ltd., Seoul, Korea) were used. To assist the animals in adapting to their environment, all rabbits were housed in individual cages with freely available food and water for 1–2 weeks, according to existing human–animal care protocols. All animals were anesthetized intramuscularly before experiments with 5 mg/kg of zylazine (Rompun, Bayer Korea Inc., Seoul, Korea) and 5 mg/kg of alfaxalone (Alfaxan, Jurox Pty Ltd., Hunter Valley, NSW, Australia).

#### 2.3.1. Mouse Subcutaneous Tumor Model

Lewis lung carcinoma (LLC) cells were used to establish the subcutaneous tumor in the mice. For in vivo experiments, LLC cells (20 μL of 2 × 10^6^ cells/mL) were injected subcutaneously into the hind legs. The tumor model was established after 2–3 weeks. ICG (5 mg/kg; Daiichi-Sankyo Co., Tokyo, Japan) was injected into the tail vein. The fluorescence signal was observed using the ARFI system 12 h after injection.

#### 2.3.2. Rabbit Lung Cancer Model

A rabbit lung cancer model was established as per a previous study [35]. Briefly, VX2 single-cell suspensions with Matrigel solutions were directly injected into the rabbit’s lung using a 28-gauge needle. The in vivo experiment was performed after two weeks. The VX2 model of the lung tumor was established 2–3 weeks after administration. ICG (5 mg/kg) was injected into the ear vein, and the fluorescence signal was observed using the ARFI system 12 h after injection. Normal rabbits were used for the detection of the intersegmental line. After ligating the right middle lobar pulmonary arteries and vein in each rabbit, 0.6 mg/kg of ICG (*n* = 3 for each group) was injected into the ear vein. The fluorescence signal was observed using the ARFI system after injection.

## 3. Results

### 3.1. System Evaluation

FOV tests and image-matching ratio tests were conducted to evaluate the performance of the proposed system. First, as shown in Figure 3a, an ICG of 128 μM was prepared and used in a cotton swab. The light source for exciting the ICG was prepared by attaching a 790 nm short-pass filter (FF01-790/SP-25; Semrock, Rochester, NY, USA) at the front of an LED source (M780L3, Thorlabs, Newton, NJ, USA) with a 780 nm peak. Figure 3b shows the results of the FOV test. The blue rectangles denote the FOV of the system, and the green light represents the fluorescence signal. The results showed that the system provides consistent matching images, which means that the fluorescence image is always matched on the IGG sample, even when the sample is moved. This indicates that the direction in which the user wears the system and moves their head does not influence the performance of the system.

Additionally, as Figure 4 shows, we calculated the matching ratio, which indicated how closely the real-field and fluorescence images were matched. First, the fluorescence signal was changed to orange to increase the contrast with the green ICG sample. Subsequently, to calculate the matching percentage, the two images were cropped to the same size in MATLAB (Mathworks, Natick, MA, USA) and converted to grayscale. Thereafter, the matching percentage was calculated using the corr2 function. We calculated the percentage five times in the system’s FOV and averaged the obtained values. The result showed an approximately 95.3% match. Each pixel intensity above the threshold was processed to have a value of 1 to ensure that the difference between the real part and the fluorescence image could be easily interpreted.

### 3.2. Tumor Detection Using Arfi System in Mouse Tumor Model

The mice had a subcutaneous tumor in the left thigh, with a mean tumor diameter of 0.5 ± 0.2 cm (range 0.4–0.6 cm). The tumor was successfully detected through the NIR fluorescence signal in all mouse models, as presented in Figure 5. In addition, only tumor and injection sites are highlighted in the ARFI system but invisible in the surrounding scene, which indicates that the ARFI system can identify tumor margins during a surgical procedure.

### 3.3. Tumor Detection Using Arfi System in Rabbit Lung Tumor Model

The rabbit lung tumor model was successfully established in all four rabbits. The mean tumor diameter was 0.8 ± 0.3 cm. In all the models, we successfully detected the lung tumor through the NIR fluorescence signal. As shown in Figure 6, in the in vivo and ex vivo images captured by the ARFI system, the NIR fluorescence matched lung tumor sites.

Furthermore, we clarified whether ARFI could evaluate the blood flow distribution in the lung lobe in real time, as shown in Figure 7. When ICG was injected intravenously, the resection area of the lung lobe where the blood vessel was ligated was easily distinguishable using ARFI. Consequently, a surgeon can easily resect the exact area.

## 4. Discussion

This study was conducted to develop an intraoperative fluorescence imaging system using the AR technique and validate the performance of this system through laboratory-level and preclinical studies. The proposed ARFI system was developed to address the FOV difference and inconvenience associated with existing fluorescence navigation systems. We devised a new optical system to solve these problems; this system can exactly match the fluorescence pathways with the visible light. Consequently, the user can directly view the surgical field through fluorescence images, without having to look at a computer display. Furthermore, the proposed ARFI system can guide complete tumor resection in mice and rabbits.

One of the main advantages of the proposed system is that it avoids complicated image-matching algorithms. Previous studies on see-through HMD systems have used two cameras to separate the visible light and NIR fluorescence light [29,36,37]. Therefore, the camera angles of the two separately acquired images could not be directly aligned. If this issue is not addressed, the projected fluorescence image could cause severe eye fatigue as the user would need to switch focus between the real image and the fluorescence image. Therefore, an additional image-processing algorithm was needed [38]. However, our system can be more consistent with fluorescence images to the actual area using the mount which can more precisely align, and simple algorithms which can adjust the size or rotate the angles of the supplied fluorescence images. Furthermore, the image remains aligned even when the wearer moves or the glass is tilted. This advantage has been confirmed not only at the laboratory level but also in various preclinical studies, such as mouse tumor models and rabbit lung tumor models (Appendix A). The pre-use calibration feature for the eye positions of different individuals provides consistent images during use, which ensures the stability required for use in an actual operating room. This feature potentially allows for a quicker and safer procedure and could consequently enhance patient safety.

Despite the achievements, however, this study has some limitations. Although the effectiveness of tumor detection in animal models was confirmed, accurate detection of deep lesions has not yet been achieved. In addition, ICG can only be detected up to 1 cm below the skin. Therefore, certain recording positions may not be usable [39]. In this study, we detected steady-state fluorescence using a CMOS camera. However, steady-state fluorescence intensity has some limitations such as photobleaching of the NIR fluorophore, phototoxicity, and changes in fluorescence collection [40]. Therefore, for sensitive detection of tumors, it is expected that such a high fluence rate will be required. Additionally, through system evaluation, it was confirmed that the ARFI system has a high image-matching ratio. However, since the matching ratio between visible and fluorescence images can vary depending on the ICG concentration and fluence rate, the optimal ICG concentration and fluence rate are required to obtain an accurate fluorescence image during the surgery (Appendix A). Moreover, since our system does not have a visible-fluorescence image alignment algorithm, the visible and fluorescence images were manually matched. Therefore, a slight error inevitably occurs, and an algorithm that can match two images is additionally required for a perfect correlation. Although the tumor was confirmed through biopsy, tumor margin estimation should be considered for minimal incision and minimally invasive surgery [41].

Furthermore, the current system uses ready-made see-through HMD primarily designed for entertainment purposes. Although this type of display is useful for demonstrating the feasibility, we will develop a customized and ergonomic display unit optimized for medical imaging.

## Figures and Tables

**Figure 1 diagnostics-11-00927-f001:**
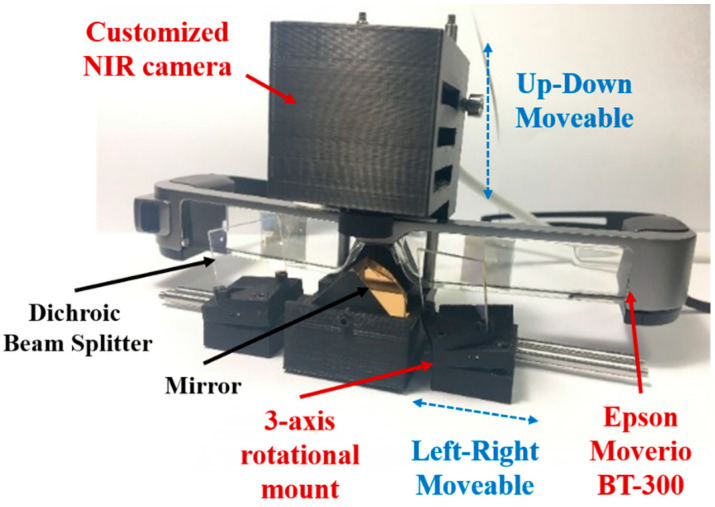
Components of the augmented reality-based fluorescence imaging goggle system.

**Figure 2 diagnostics-11-00927-f002:**
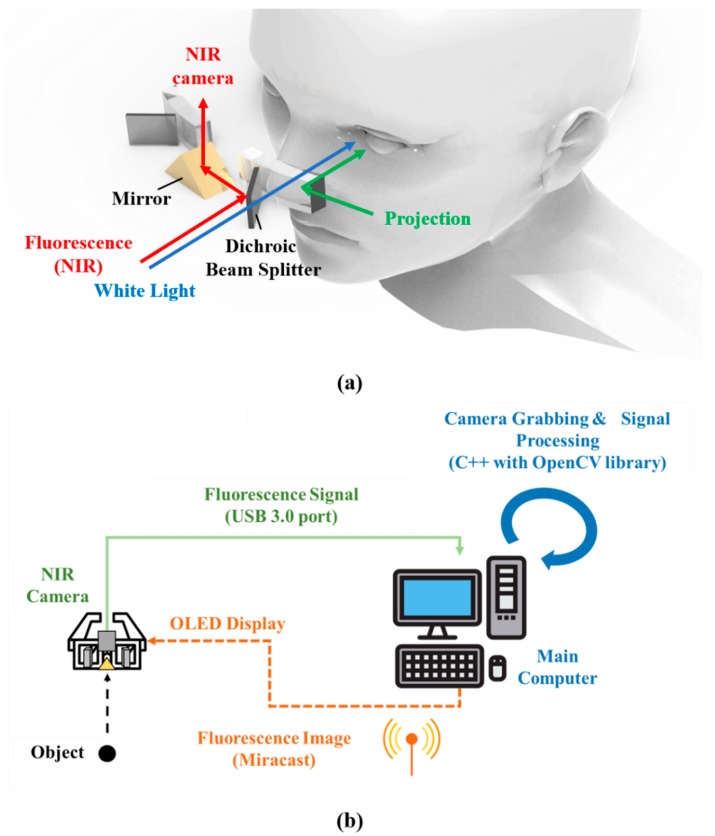
(**a**) Schematic of the ARFI system; it is characterized by matching the pathways of NIR fluorescence light with the white light that is transmitted to the wearer’s eyes. (**b**) Operation flow of the entire system.

**Figure 3 diagnostics-11-00927-f003:**
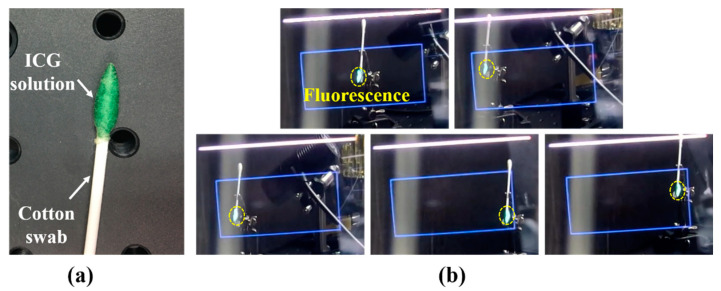
(**a**) ICG sample in cotton swab head, (**b**) FOV test result; blue rectangle is the FOV of the system, and green light is the fluorescence signal. In the blue rectangle, the fluorescence signal is always matched on the ICG sample.

**Figure 4 diagnostics-11-00927-f004:**
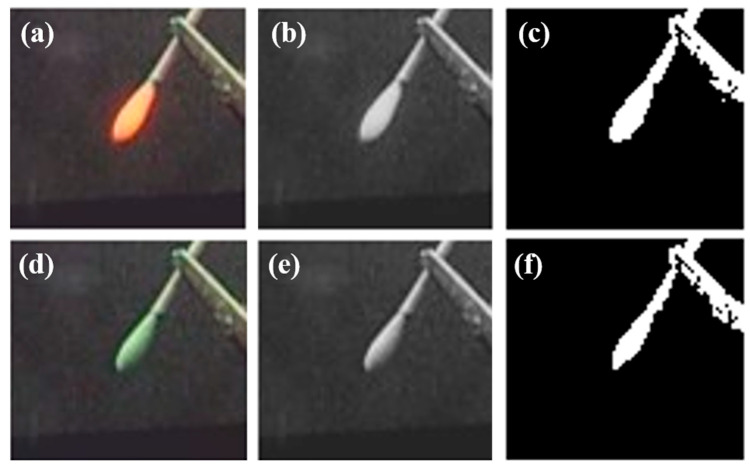
Process of calculation the matching percentage of fluorescence image. (**a**) Cropped image of ICG sample, (**b**) gray scale image of ICG sample, (**c**) binarize of ICG sample image. Above the threshold, all pixel intensities were 1. (**d**) Cropped fluorescence image, (**e**) grayscale image of fluorescence, and (**f**) binarized fluorescence image. Above the threshold, all pixel intensities were 1.

**Figure 5 diagnostics-11-00927-f005:**
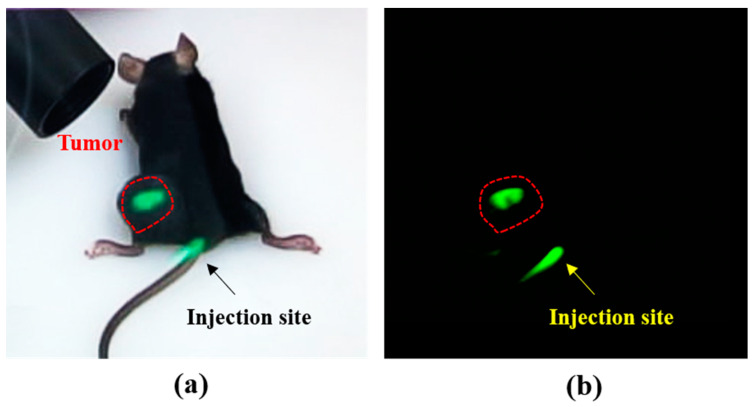
Subcutaneous tumors in the mice detected by the ARFI system. (**a**) Representative real image of the mouse model observed using the ARFI system, and (**b**) representative fluorescent image of the mouse model observed using the ARFI system.

**Figure 6 diagnostics-11-00927-f006:**
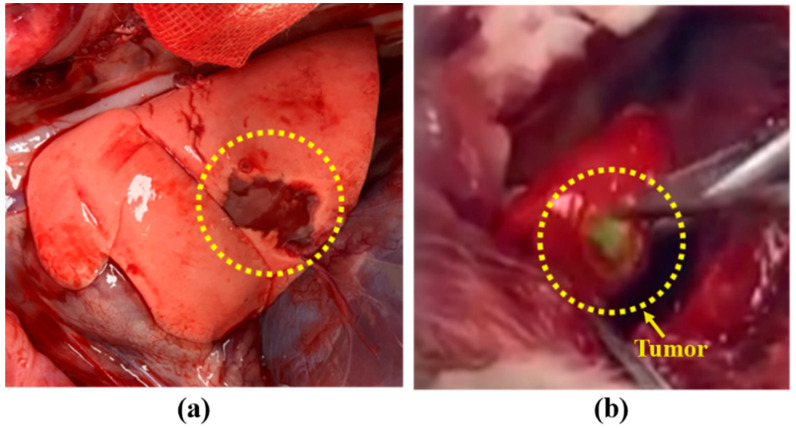
Lung tumor in rabbit detected by the ARFI system. (**a**) Representative color image of rabbit lung tumor, (**b**) representative real image of lung tumor observed using the ARFI system.

**Figure 7 diagnostics-11-00927-f007:**
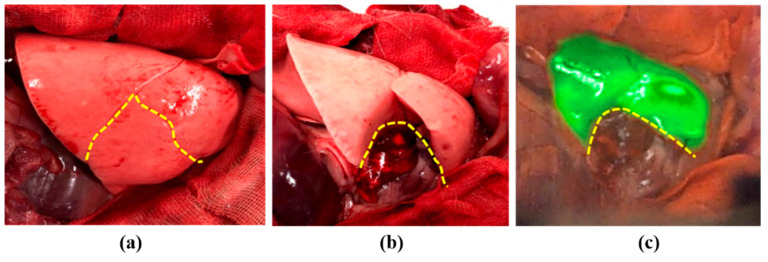
Distribution of blood flow in the lung lobes detected by the ARFI system. (**a**) Color image of rabbit lung before ligation of lobar pulmonary vein and artery, (**b**) color image of rabbit lung after ligation of lobar pulmonary vein and artery, (**c**) real image of blood flow distribution in the lung lobes observed using ARFI system; yellow dashed line indicates the pulmonary lobe of the ligated lobar vein and artery.

**Table 1 diagnostics-11-00927-t001:** Overview of wearable fluorescence imaging systems.

Authors	Display Module	Hardware Design	Image	Application
Y. Liu et al. 2011 [21]	Monocular night vision viewer	Combined Night vision viewer, White/NIR light source	Monochrome/fluorescence fusion image	SLN mapping (preclinical) HCC imaging (clinical)
Y. Liu et al. 2013 [32]	Binocular HMD (ST1080, Silicon microdisplay)	Combined HMD, CMOS Camera, NIR light source	Monochrome/fluorescence fusion image or natural vision	SLN mapping Liver cancer surgery (preclinical)
P. Shao et al. 2014 [27]	Monocular HMD (Google glass, Google Labs) Binocular HMD (Personal Cinema System, Headplay)	Non-combined H.M.D., CCD. camera, NIR light source Combined HMD, CMOS camera Non-combined C.C.D. camera, NIR light source	Fluorescence image superimposed on natural vision Color/fluorescence fusion image	Phantom study
Mela CA et al. 2015 [33]	Binocular HMD (lab made)	Combined HMD, four CMOS sensors Non-combined hand-held microscopy, NIR light, ultrasound scanner	Color/fluorescence fusion image	Phantom study
S. B. Mondal et al. 2015 [24]	Binocular HMD (Carl Zeiss)	Combined HMD, custom VIS-NIR camera Non-combined NIR light	Color/fluorescence fusion image	Ovarian cancer surgery (preclinical) SLN mapping (clinical)
Zhang Z et al. 2016 [28]	Monocular HMD (Google glass, Google Labs)	Non-combined H.M.D., CCD. camera, NIR light, ultrasound probe	Fluorescence image superimposed on natural vision	SLN mapping (clinical)
S. B. Mondal et al. 2017 [29]	Binocular HMD (Carl Zeiss)	Combined HMD, custom VIS-NIR camera Non-combined NIR light	Color/fluorescence fusion image	Tumor resection (preclinical), SLN biopsy (clinical)
M. Keisuke et al. [30]	Binocular HMD (Moverio BT-200, Epson)	Combined HMD, optical markers Non-combined motion capture cameras	Fluorescence image superimposed on natural vision	Brain tumors
Our smart goggle system	Binocular HMD (Moverio BT-300, Epson)	Combined HMD, CMOS camera, and optical system Non-combined NIR light source	Fluorescence image superimposed on natural vision	Cancer detection, segmental line identification (preclinical)

## Data Availability

Data sharing not applicable.

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
