# Peer review of "Design and Testing of Augmented Reality-Based Fluorescence Imaging Goggle for Intraoperative Imaging-Guided Surgery"

_diagnostics, 2021, doi:10.3390/diagnostics11060927_

Round 1
Reviewer 1 Report
The authors have clearly described and developed a useful additional capability in a goggle for tumour monitoring. The dual monitoring of white light and ICG fluorescence provides a real time capability that has significant advantages over single channel other approaches. The optical system to achieve this is nicely designed.
However, the paper would benefit from inclusion of some relevant aspects to the “bigger picture” of fluorescence and tumour observation:
1. The overlay of different fluorescence and white light image has been described previously and reviewed. The relevance of prior work should be included in the Introduction and comparisons with the present work discussed. See for example:
Troyan S L et al The FLARE intraoperative near-infrared fluorescence imaging system: a first-in-human clinical trial in breast cancer sentinel lymph node mapping. Ann. Surg. Oncol., (2009) 16 (10), 2943
D’Souza A V et al Review of fluorescence guided surgery systems: identification of key performance capabilities beyond indocyanine green imaging J. Biomed. Opt. (2016) 21 080901-1-15
2. Tumour margin estimation is central to the aims of this type of study and yet it is not discussed. The paper would be greatly improved by some consideration of this including actual data to support the conclusions. The animal models are useful at the level considered but estimation of the tumour margin for both white light and ICG fluorescence in both these cases would be very informative and indeed is necessary.
3. The study uses a CMOS camera to detect steady-state fluorescence. The limitation of using steady-state fluorescence intensity need to be spelled out in the paper. For example dye bleaching for which ICG is well known, effect of dye concentration variations, etc are very relevant and should not be ignored but included.
4. Also in respect of points 2 and 3 the use of fluorescence lifetime imaging in tumour margin observation is starting to become a credible possibility as it overcomes the limitations of fluorescence intensity monitoring. The development described could actually benefit lifetime imaging also. Hence to bring the paper more up to date with broader developments in the field some consideration with recent publication citations would be a useful addition to either the Introduction or Conclusion. Recent papers of relevance include:
Homulle H A R et al Compact solid-state CMOS single-photon detector array for in vivo NIR fluorescence lifetime oncology measurements Biomed. Opt. Express (2016) 7 1797
Stewart H L et al Characterization of single channel liquid light guide coupling and SPAD array imaging for tumour margin estimation using fluorescence lifetime
Meas. Sci. Technol. (2020) 31 125701
Author Response
Dear Editor and Reviewers,
We would like to thank you and the reviewers for your comments on our manuscript “Design and Testing of Augmented Reality-based Fluorescence Imaging Goggle for Intraoperative Imaging-guided Surgery”. We have revised the manuscript to address the reviewers’ concerns and attached a point-by-point response to each of the reviewers’ comments below.
We greatly appreciate the opportunity to have our manuscript reviewed and considered for publication in Diagnostics. We hope that the revised manuscript is now more appropriate to the readership of your esteemed journal.
We look forward to hearing from you.
Yours sincerely,
Beop-Min Kim
Department of Bio-convergence Engineering, Korea University, 145 Anam-ro, Seoul, South Korea, 02841
Tel: +82-2-940-2771
Fax: +82-2-941-2745
Email: bmk515@korea.ac.kr

Reviewer 2 Report
The authors present an AR goggle system for fluorescence intraoperative guidance. The paper is well presented and the topic quite interesting. However, the reviewer has some concerns regarding the viability of such technology.
- In many cases surgeons wear protective goggles. How would the proposed goggles interfere with those?
- How about surgeons that wear glasses?
- How about sterilization?
- What is the weight of the proposed googles?
- How stable are the goggles in relation to head movement? In many cases rapid head movements are to be expected during surgery. What is the risk of the goggles falling onto the patient?
- Is the working distance strictly fixed? What is the the depth of view and is there any algorithmic approach to detect when the system is out of focus so the stop/pause the projection?
More over the reviewer would expect to see a more extended performance assessment of the proposed goggles. For example, what is their dynamic range as a function of ICG concentration or which is the resolution of the system?
Finally, in some cases in the results information already presented is repeated. For example the beginning of section 3.1 was already presented in section 2.1.
Author Response

(The authors gave the same response as above.)

Reviewer 3 Report
Developed in the mid-50’s, the fluorescing dye Indocyanine Green Angiography (ICG) has grown interest for the past decades in biomedical/clinical imaging applications. This radiation-free simple and safe dye are now routinely used in medicine thanks partly to the continuous computing/engineering development efforts. Due to its spectral property, much emphasis has been put to optimize near-infrared imaging devices. While the architecture can dramatically differ, the quality of the following core components defines the performance of the device: the light source, the excitation/emission optical path components and the chip sensor to collect emission signal. In this work, Lee and colleagues showcased a compact device termed ARFI goggle as a novel and promising intraoperative system. While the optical design is sound, few clarifications are needed.
Find below my comments:
- While the description of the optical components is well detailed, the authors failed to report the processing step applied to acquired NIR images. Could the authors elaborate on the “post-processing” (line 128)? Is the post-processing applied with respect to the integration time (60 frames/second, line 83) of the CMOS camera? the fluence rates?
- Despite being a reliable dye, ICG still suffers from a low quantum yield (Sevick-Muraca et al., 2011) that could either be compensated with sensitive CCD-based detectors or increasing the incident light (at a cost of higher background or heating the zone of interest). The result provided in section 3.1 could be strengthen by varying the concentration of ICG: how is the visible-IR FOV correlation changing with decreased collection signal? Integration time? Fluence rates (optical density filters could be used for variation if the LED source could not be tuned, line 110)?
Similarly how could the authors explain the 5% gap to reach the perfect correlation (95.3% in line 191)? Is it dependent of the size/geometry of the observed/dyed objects? The integration time? Or simply due to the segmentation threshold method?
Such information would be crucial to assess the performance of such device.
- The authors should tone down the sentence in line 203 due to the margin of error they determined in line 191 as well as the lack of information in depth (3D).
- Could the author explain the discrepancy between the panels a and b in figure 6? Figure 6a shows a much bigger display of the tumor region compared to the IR image. Is it due to the low Signal to Noise ratio? The concentration of ICG? The inherent depth-limitation of such imaging?
- In line 245, the authors mentioned that the ARFI system could tailor the fluorescent images via hardware adjustments (those mentioned in lines 95-97?) combined to registration algorithms (line 246). This is not clear.
Right after manual adjustments mentioned in lines 95-97, this is followed by software-based adjustments (line 246). If so, what is the reference for the adjustments?
- The authors could provide with an estimated cost of this device. This would help to relate to other existing devices.
Author Response

(The authors gave the same response as above.)

Round 2
Reviewer 1 Report
The authors have considered the review comments and responded in a reasonable way. The area is one of growing interest and the paper makes a contribution to this through the optical design in particular and thus is worthy of publication.
Reviewer 2 Report
Based on the feedback from the authors, the reviewer thinks that the submitted paper describes an incomplete system. The reviewer suggests the authors to reconsider submitting the manuscript once the goggles are finalized.